# Near-Surface-Defect Detection in Countersunk Head Riveted Joints Based on High-Frequency EMAT

**DOI:** 10.3390/ma16113998

**Published:** 2023-05-26

**Authors:** Shuchang Zhang, Jiang Xu, Xin Yang, Hui Lin

**Affiliations:** 1School of Mechanical Science and Engineering, Huazhong University of Science and Technology, Wuhan 430074, China; sczhang@hust.edu.cn (S.Z.); yangxin_@hust.edu.cn (X.Y.); 2Huitong Construction and Installation Engineering Company, Daqing 163000, China; dqhtgs@126.com

**Keywords:** countersunk head riveted joint, defect, high frequency, EMAT

## Abstract

Countersunk head riveted joints (CHRJs) are essential for the aerospace and marine industries. Due to the stress concentration, defects may be generated near the lower boundary of the countersunk head parts of CHRJs and require testing. In this paper, the near-surface defect in a CHRJ was detected based on high-frequency electromagnetic acoustic transducers (EMATs). The propagation of ultrasonic waves in the CHRJ with a defect was analyzed based on the theory of reflection and transmission. A finite element simulation was used to study the effect of the near-surface defect on the ultrasonic energy distribution in the CHRJ. The simulation results revealed that the second defect echo can be utilized for defect detection. The positive correlation between the reflection coefficient and the defect depth was obtained from the simulation results. To validate the relation, CHRJ samples with varying defect depths were tested using a 10-MHz EMAT. The experimental signals were denoised using wavelet-threshold denoising to improve the signal-to-noise ratio. The experimental results demonstrated a linearly positive correlation between the reflection coefficient and the defect depth. The results further showed that high-frequency EMATs can be employed for the detection of near-surface defects in CHRJs.

## 1. Introduction

Countersunk head riveted joints (CHRJ) are widely used in the aerospace, marine, and mechanical manufacturing industries. Stress concentrations are common in the joint section, and they may lead to defects [1,2,3]. Countersunk head riveted joints can be subjected to significant stresses, resulting in potential defects near the lower boundary of the countersunk area due to stress concentrations. The detecting and identification of defects in CHRJs are crucial to ensure safety. Over the years, scholars have extensively researched non-destructive testing (NDT) methods for CHRJs. Pulsed eddy current technology (PECT) is broadly used to detect defects in aluminum plates in CHRJs. Sophian et al. [4] proposed a new pulsed-eddy-current probe and circuit system and developed a compensation technique to improve the signal resolution. Abidin et al. [5] proposed an optimized inspection method for riveted joints, while He et al. [6] designed three different pulsed-eddy-current probes to detect defects in aluminum plates in riveted joints. In addition to PECT, Sieber et al. [7] used locked thermography to detect defects around the hole in the plate in riveted joints, while Fromme et al. [8] employed a scattering field of lamb waves to detect defects in aluminum plates around rivet holes. However, these inspection methods are difficult to use for the inspection of rivets in CHRJs [9]. Some scholars have utilized ultrasonic testing (UT) technology for the detection of defects in rivets in CHRJs. Le et al. [9] and Amosov et al. [10] used the ultrasonic-pulse-echo method with piezoelectric transducers (PZTs) for defect detection in rivets. However, all these scholars used piezoelectric transducers for detection. Although PZTs offers advantage such as high detection accuracy and transducer efficiency, conventional PZT-based ultrasonic testing is inefficient and requires a coupler. Additionally, laser ultrasonic testing requires larger equipment and may not be suitable for certain applications. Therefore, it is imperative to explore detection methods based on electromagnetic acoustic transducers (EMATs) for defects in CHRJ rivets.

The EMAT method is widely used for detecting defects in conductive materials. Hirao et al. [11] introduced the principle of the EMAT and proposed the EMAT structure for various common detection objects. Parra-Raad et al. [12] used an orthogonal butterfly EMAT to detect defects in metals. Petcher et al. [13] utilized the SH wave generated by periodic permanent magnet (PPM) to detect defects in weld joints and found that SH-wave PPM–EMAT is more efficient than PZTs. Park et al. [14] analyzed the second harmonic amplitude at different locations in a steel wire to identify nonlinear defects. He et al. [15] utilized Rayleigh waves generated by EMATs to detect the depths of defects on metal surfaces. Parra-Raad et al. [16] detected cracks and anisotropy in metals using transverse waves generated by two orthogonal butterfly-coil EMATs. Li et al. [17] applied a spiral-coil EMAT to inspect subsurface cracks in rails. Andruschak et al. [18] used PPM–EMAT to detect defects in pipe supports and identified the region on the dispersion curve that is sensitive to defects. Based on the above research, EMATs can be used to detect defects in metals. Furthermore, EMATs can also adapt to different detection requirements by changing its structure. Despite the success of EMATs in defect detection in the studies described above, the detection of near-surface defects in CHRJs presents several challenges. First, CHRJs and the defects in them are small in size, which results in a small ultrasonic signal amplitude and a low signal-to-noise ratio (SNR). Second, defects located near the surface are demanding on the axial resolution. Third, near-surface defects cannot be detected directly by the signal near the initial pulse due to the influence of the ultrasonic signal’s blind zone. The blind zone of the ultrasonic signal is the part of the signal that cannot be used for defect detection in the initial part of the signal, which is usually manifested as a direct-current level. The blind zones in ultrasonic signals are generated by hardware and cannot be avoided, so their impact needs to be considered during detection. For the problem of low SNR, many scholars have studied the denoising method for analog signals. Si et al. [19] utilized variational mode decomposition (VMD) and wavelet transform (WT) to reduce the noise in large extracted EMAT signals. Huang et al. [20] proposed a mode-identification method using the VMD algorithm and a time-of-flight (TOF) extraction method. Zhao et al. [21] proposed a least-mean-square adaptive-filtering-interpolation-denoising method based on VMD and wavelet-threshold denoising for high-temperature EMAT signals. Kubinyi et al. [22] developed a filtering method for EMAT signals based on stationary wavelet packet denoising with a threshold determined by the signal characteristics. Li et al. [23] developed a wavelet-packet-analysis algorithm to effectively detect grouted defects using the ultrasonic method. Huang et al. [24] proposed a denoising method based on the envelope of the original signal. The above research on signal denoising has effectively improved the SNR. For the EMAT-detection signal in CHRJs, the denoising method should be selected according to the characteristics of the signal. For the characteristics of defects near the surface in CHRJs, a high-frequency transverse-wave EMAT should be chosen to ensure axial resolution. Meanwhile, it is necessary to select the signal after the first bottom echo for detection in order to avoid the effect of the ultrasonic detection signal’s blind zone. This study is based on the propagation characteristics of elastic waves in a CHRJ from a previous study [25].

In this paper, a near-surface defect in the CHRJ was detected based on a high-frequency EMAT. The originality of this paper is mainly demonstrated by the following points: The propagation of ultrasonic waves in a CHRJ with a defect is analyzed using the theory of the reflection and transmission of ultrasonic waves.The principle of defect-echo generation is studied using the FES, and the relationship between the defect depth and the reflection coefficient is investigated.An experimental verification of the simulation results was performed, the experimental signal was processed using the WTD, and the simulation results were in good agreement with the experimental results.

## 2. Theory

### 2.1. Propagation of Ultrasonic Waves in the CHRJ with a Defect

The structure of the CHRJ and the location of the potential defect are illustrated in Figure 1. As this paper focuses on detecting the defect using the signal after the first bottom echo, the impact of defects on ultrasonic wave propagation is intricate. Thus, determining the depth of the defect solely based on the amplitude of the defect signal is not feasible, and a thorough analysis of ultrasonic wave propagation in the CHRJ is required. Figure 2 depicts the propagation of ultrasonic waves in the CHRJ with a defect, based on an analysis of the reflection-and-transmission theory of ultrasonic waves in defects.

The reflection and transmission occur each time the ultrasonic waves pass through the defect, thus necessitating the introduction of transmission and reflection coefficients to describe the energy transfer. Additionally, given the propagation characteristics of ultrasonic waves in CHRJs, part of the energy enters the aluminum plate or is lost in the upsetting part [25]. Hence, it is essential to introduce the energy-loss coefficient, denoted as a. The defect-signal energy utilized in this paper refers to $art^{2}E$. To characterize the defect depth using the defect-signal amplitude, the transmission coefficient needs to be considered. The transmission coefficient cannot be calculated because the signal can only be received at the upper surface of the rivet, so the effects of *t* need to be excluded. The reflection coefficient of the ultrasonic wave in the defect can effectively characterize the depth of the defect and exclude the effect of t. According to the signal characteristic, *r* can be expressed by Equation (1). The definitions of the variables in Figure 2 and Equation (1) are shown in Table 1.
(1)$$r=\frac{art^{2}E}{at^{2}E}=\frac{A_{De}}{A_{Fbe}}$$

### 2.2. High-Frequency Transverse-Wave EMAT for the CHRJ

The defect in the CHRJ is close to the upper surface, so the pulse width needs to be minimized in order to improve the axial resolution. In order to identify defect echoes in the signal, the axial resolution must be less than the distance from the defect to the upper surface, and the axial resolution is not related to defect size. The axial resolution of pulse-echo ultrasonic testing is shown in Equation (2). The definitions of the variables in Equation (2) are shown in Table 2.
(2)$$d_{ax}=\frac{nc}{2f}<h$$

To minimize the $d_{ax}$, the transverse wave is used for detection as the transverse-wave speed in aluminum is about half the speed of the longitudinal wave. Due to the small size of the CHRJ, the number of cycles needs to be increased in order to increase the energy of the ultrasonic waves. When the n is three and for near-surface defects in the CHRJ, the frequency needs to be at least greater than 3 MHz to ensure axial resolution. Since the exciting of ultrasonic waves may lead to an increase in pulse width, a higher frequency is required to ensure the feasibility of defect detection. In this paper, the 10-MHz transverse wave is chosen and in this case, the $d_{ax}=0.48\;\mathrm{mm}$, so the 10-MHz transverse wave can meet the requirement of axial resolution.

The schematic diagram of transverse-wave EMAT is shown in Figure 3. The transverse-wave EMAT is composed of a magnet, coil, and copper backing plate. When exciting ultrasonic waves, a high-frequency current is generated in the coil. According to Faraday’s law, induced eddy currents are generated in the conductor near the coil. The interaction of this eddy current with the static magnetic field of the magnet produces ultrasonic waves. The reception of ultrasonic waves can be accomplished by reversing this process. The function of the copper backplate is to prevent the generation of ultrasonic waves in the magnet from affecting the detection results. The basic equation for ultrasonic excitement and reception in EMAT is shown in Equation (3). The definitions of the variables in Equation (3) are shown in Table 3.
(3)$$\left\{\begin{array}{c} F_{lor}=J_{ind}\times B_{static}\\ J_{re}=\eta \left(v\times B_{static}\right) \end{array}\right.$$

## 3. FES of Defect Detection

### 3.1. Modeling

To investigate the effect of defects on ultrasonic wave propagation in CHRJ, the simulation model of the detection process is established by the commercial software, COMSOL Multiphysics. The design of the simulation model is based on a previous study [25].

Since the CHRJ is a rotationally symmetric structure, the 2D module is used for modeling in order to simplify the model. The geometric model of the simulation model is shown in Figure 4. To excite transverse waves in the CHRJ, the “boundary load” module is used to apply horizontal force in the CHRJ. The contact surface between the aluminum plate and the rivet is set as the “elastic thin layer” in the model. The maximum mesh size in the model is 0.05 mm and the minimum mesh size is$\;2.8\times 10^{-4}$ mm. In the simulation of defects, the defect is set as a cavity in the mesh and the material is set to air. In order to investigate the relationship between defect depth and signal characteristics, the CHRJ was simulated for different defect depths. The width of defect is 0.2 mm, and the depths of defect are 0.5 mm, 1 mm, 1.5 mm, and 2 mm, respectively.

### 3.2. Simulation Results

To investigate the impact of defects on ultrasonic wave propagation, it is imperative to analyze the energy distribution of ultrasonic waves in CHRJ. Figure 5 displays the kinetic energy-density distribution in the CHRJ with a 0.5-mm defect at 10.285 μs. At this point, the ultrasonic waves have undergone one reflection on the lower surface and one reflection on the upper surface of the rivet, and have passed through the defect three times. The defect echo in Figure 5 corresponds to the reflection echo when the main transverse wave passes through the defect for the third time. Detection of the defect can be achieved by identifying the defect echo on the upper surface.

The receiving signals on the upper surface are shown in Figure 6. The *x*–axis is converted to distance according to the transverse-wave speed. For the defect-free CHRJ, the receiving signal is relatively pure. The signal at about 7 mm is the reflection echo of the conical surface of the rivet countersunk head. For the detection signal of CHRJ with a 0.5-mm defect depth, the major difference is the presence of the defect echoes. The defect echo S1 appears after the initial signal, but cannot be utilized for defect detection as it falls within the ultrasonic detection signal’s blind zone. On the other hand, the defect echo S2 remains unaffected by the ultrasonic signal’s blind zone and can be employed for defect identification, which corresponds to the defect echo in Figure 5. To investigate the effect of defect depth on the receiving signal, simulations were conducted for defect depths of 0.5 mm, 1 mm, 1.5 mm, and 2 mm in CHRJ. The results of the simulations are presented in Section 4.3.

## 4. Experiments

### 4.1. Experiment Setup

To verify the simulation results, experiments were performed on a CHRJ sample. The arrangement of the experiment is shown in Figure 7. The electronic system of the experiment mainly consists of three parts: power amplifier, weak signal amplifier, and oscilloscope. The role of the power amplifier is to provide high-voltage pulses to the EMAT to excite the ultrasonic waves in the CHRJ. The role of the weak-signal amplifier is to amplify the ultrasonic signal received by the EMAT. The function of the oscilloscope is signal acquisition and display. In this experiment, the RITEC RPR4000 assumes the role of power amplifier and weak-signal amplifier. Furthermore, the type of oscilloscope is TELEDYNE LECROY wavesurfer 3024. The signal is acquired directly by the oscilloscope. The parameters in the experiment are shown in Table 4. To ensure the reliability of the experiments, CHRJs with defect depths of 0.5 mm, 1 mm, 1.5 mm, and 2 mm and a defect–free CHRJ were tested. The width of the defect was 0.2 mm. The CHRJ sample, magnet, and the spiral coil of EMAT are shown in Figure 8. The parameters of EMAT are shown in Table 5.

### 4.2. Signal Process Based on WTD

The signal after pre-processing by the high-pass filter of the CHRJ is shown in Figure 9. The initial pulse is the signal generated in the initial part due to the ultrasonic excitation. The oscillation of the duplexer is generated after the signal passes through the duplexer. The ultrasonic signals that can be clearly identified in Figure 9 are the first bottom echo and the second bottom echo. They represent the reflected echoes of ultrasonic waves at the bottom of the rivet. However, the defect signal is not clearly identifiable due to the presence of significant background noise. The close frequency of the noise and signal makes it difficult for the filter to effectively enhance the SNR. Further processing of the signal is required due to the small amplitude of the defect signal and the large background noise. The flow chart of the signal processing is illustrated in Figure 10. The pre-processed signal is decomposed using wavelets, and the wavelet coefficients of each layer obtained from the decomposition are shown in Figure 11a. At this point, the Sym6 wavelet is used to decompose the signal in seven layers as the Sym6 wavelet is similar to the signal. In order to effectively remove noise, a suitable threshold needs to be determined based on the characteristics of the signal. Layers 5–7 contain the main signal components and are low in frequency. Therefore, the thresholds for each layer are calculated separately, according to the maximum–minimum criterion and the noise structure of each layer. Furthermore, according to the results of wavelet decomposition, layers 1–4 of wavelet coefficients contain almost no signal and are highly similar to noise. Therefore, the thresholds of layers 1–4 are considered as white noise when they are calculated. The thresholds are calculated based on the structure of white noise with the maximum–minimum criterion. Next, the wavelet coefficients are processed using the hard-threshold function. The wavelet coefficients after processing by the threshold function are shown in Figure 11b. Finally, the signal is reconstructed using the denoised wavelet coefficients. The EMAT-denoised signal of the defect-free CHRJ and CHRJ with defect depths of 0.5 mm is shown in Figure 12. When comparing the original signal with the denoised signal, it can be observed that the signal-denoising method described in this section is not effective in removing the oscillation noise of the duplexer. However, since the defect echo after the first bottom echo is used for defect detection at this point, the effect of duplexer-oscillation noise can be ignored. Notably, the signal-denoising method reported in this section effectively removes the noise between the first echo and the second echo.

### 4.3. Results and Discussion

In Figure 13, the *x*–axis is converted to distance according to the transverse-wave speed, as with the simulation results. The initial pulse was generated due to the high-pass filtering of the original signal in the first step of the signal processing. According to Figure 13, the defect-free-CHRJ signal was relatively pure, which was similar to the simulation results. The defect echo was clearly identified in the denoised signal, which corresponded to the defect echo S2 in the simulation signal. The difference between the horizontal coordinates of the defect echo and the first bottom echo was about 1.5 mm, which corresponded to the location of the defect in the CHRJ. Similarly, the reflection coefficient was calculated from the experimental signal to characterize the depths of the near-surface defect. To avoid the effect of the WTD on the signal amplitude, the reflection coefficient was calculated using the signal amplitude in the original signal. The relationship between the simulation reflection coefficient, the experimental reflection coefficient, and the defect depth is shown in Figure 13. Since the units of the simulation and the experimental signals were different, the reflection coefficient was normalized according to the maximum–minimum normalization. The results show that both the simulation and the experimental reflection coefficients were positively correlated with the defect depth, but that there were some errors. The main reason for the errors is that the simulation model is two-dimensional model and features direct horizontal loading, which differs from the actual conditions of three-dimensional EMATs. Meanwhile, the interface-contact stress in the simulation model may be different from that in the actual model, which changes the attenuation of the ultrasonic waves. Therefore, the depths of near–surface defects in CHRJ should be determined based on the experimental results. Although only near-surface defects in the CHRJ were detected in this study, the proposed detection method using high-frequency EMATs can also detect defects at other locations. The shank echoes mentioned in a previous study [25] were not present in the signals reported in this paper. The main reason for this is that transverse waves were used in this study, while the propagation characteristics of longitudinal waves were studied in the previous study. The use of transverse waves leads to more energy leakage into the aluminum plate. Another reason is that the CHRJ upsetting length studied in this paper was 2.5 mm; at this point, the shank-echo amplitude is very small, and it is difficult for EMAT to receive. 

## 5. Conclusions and Future Works

In this paper, a near-surface defect in a CHRJ was detected based on a high-frequency EMAT. Compared with Le [9] and Amosov [10]’s studies using PZTs, the most significant feature of this study is the near-surface-defect detection in the CHRJ using an EMAT. Furthermore, Le and Amosov’s studies only used PZTs for defect detection, without studying the principle of defect-echo generation. This paper analyzed the propagation of ultrasonic waves in a CHRJ with a defect and used FES to study the principle of defect-echo generation. Another innovation of this article is the study of the relationship between the ultrasonic reflection coefficient and the defect depth, which was not covered in previous studies on the detection of defects in riveted joints. Additionally, the original signal was denoised using a combination of a high-pass filter and a WTD to enhance the SNR. The experimental results demonstrated a linear and positive correlation between the reflection coefficient and the defect depth. The results further showed that the detection of near-surface defects in CHRJs is feasible using high-frequency transverse-wave EMATs. This paper can also provide a reference for the detection of near-surface defects in metal structures using EMATs.

In the future, we will design and optimize the EMAT for defect detection in CHRJs, with a focus on improving the SNR. Furthermore, the influence of the defect shape on the feasibility of defect detection will be investigated.

## Figures and Tables

**Figure 1 materials-16-03998-f001:**
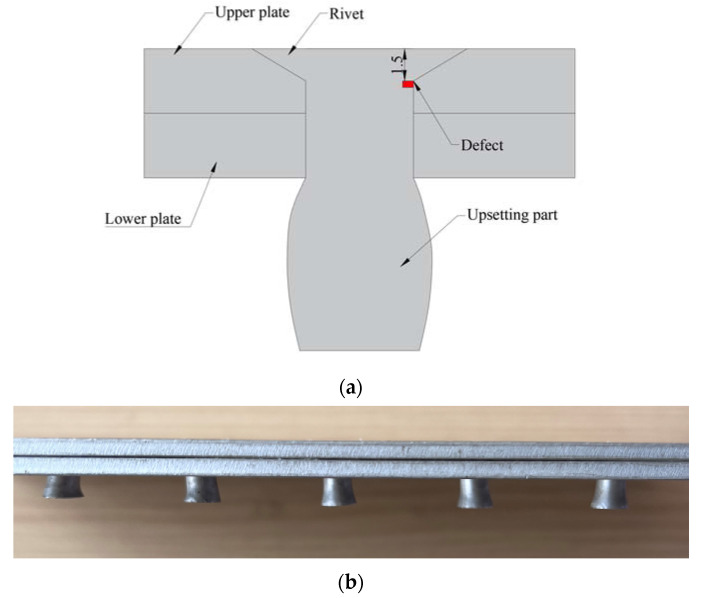
Structure of the CHRJ. (**a**) Structural diagram of the CHRJ; (**b**) photograph of the CHRJ.

**Figure 2 materials-16-03998-f002:**
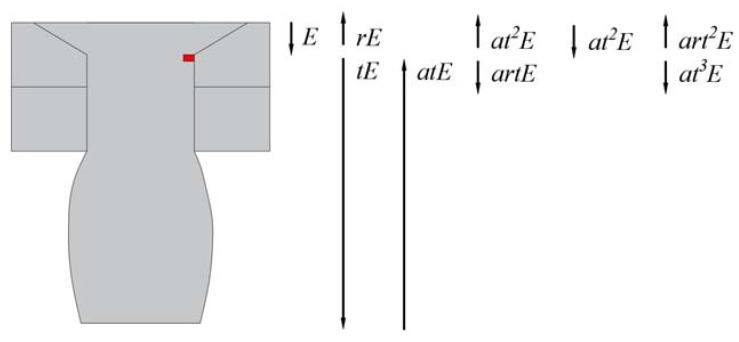
Propagation of ultrasonic waves in the CHRJ with a defect.

**Figure 3 materials-16-03998-f003:**
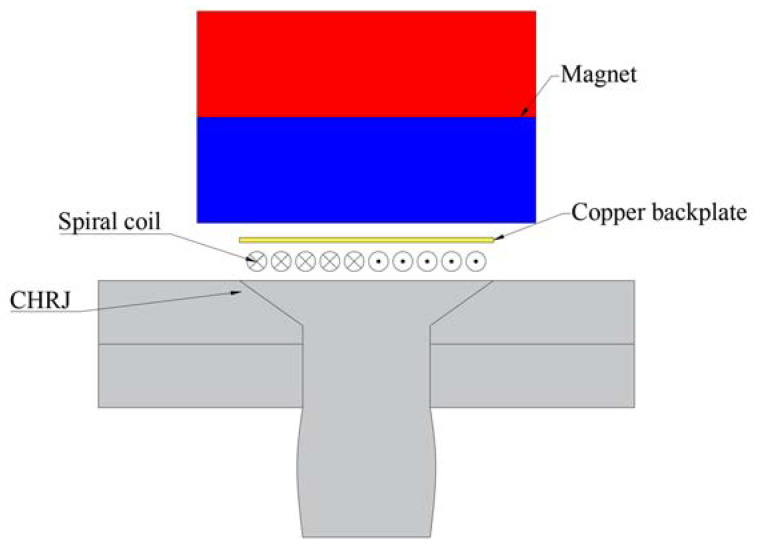
Schematic diagram of transverse-wave EMAT.

**Figure 4 materials-16-03998-f004:**
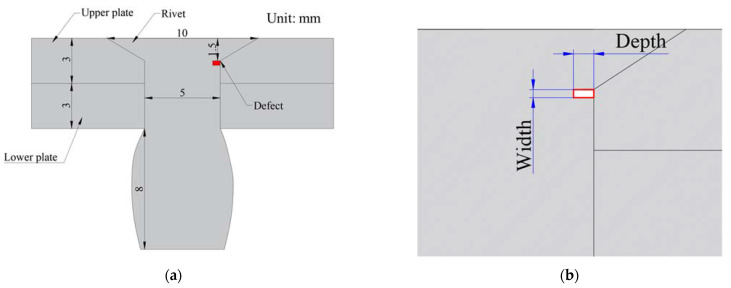
The 2D geometric models of CHRJ. (**a**) Dimensions of CHRJ; (**b**) size of the defect.

**Figure 5 materials-16-03998-f005:**
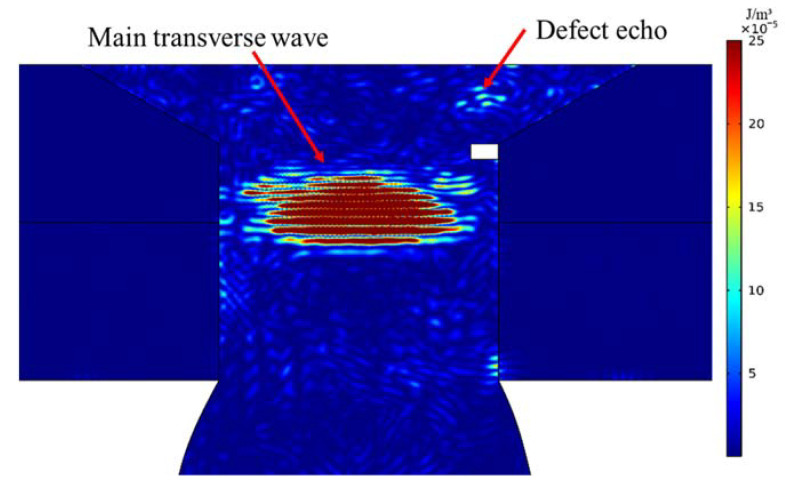
The kinetic energy field in the CHRJ with 0.5-mm defect at 10.285 μs.

**Figure 6 materials-16-03998-f006:**
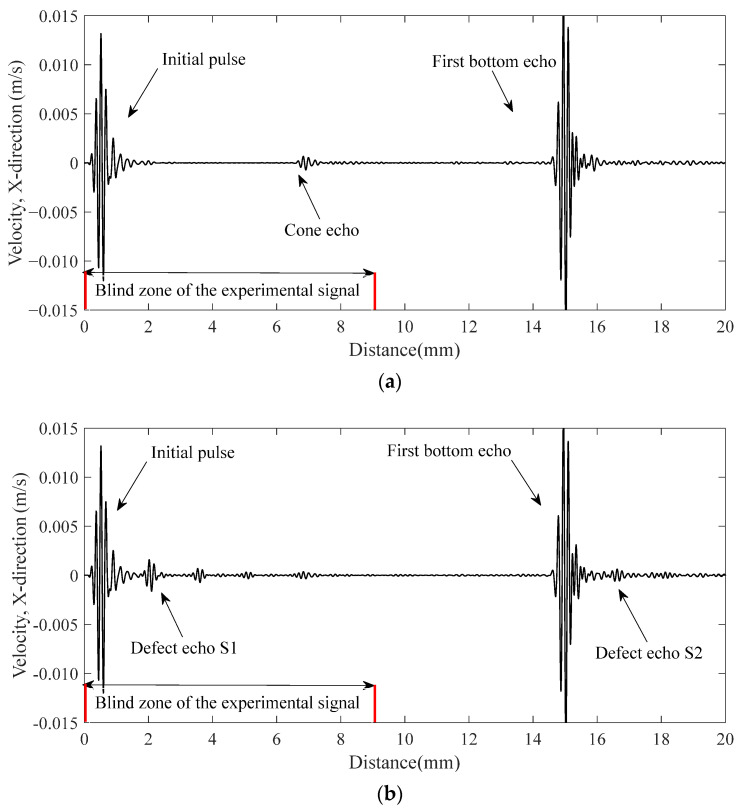
Simulation signals. (**a**) Defect-free CHRJ; (**b**) 0.5-mm-defect CHRJ.

**Figure 7 materials-16-03998-f007:**
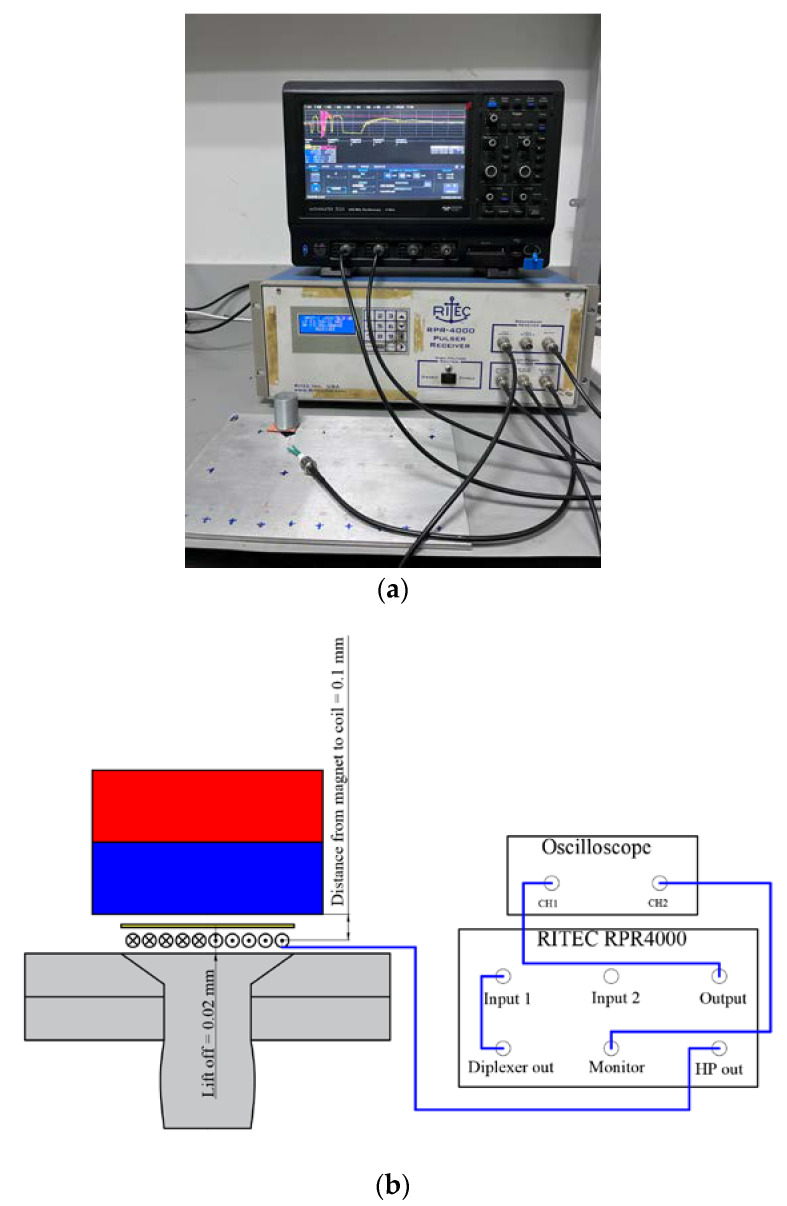
General arrangement of experiment. (**a**) Photograph of the experiment; (**b**) schematic diagram of the experiment.

**Figure 8 materials-16-03998-f008:**
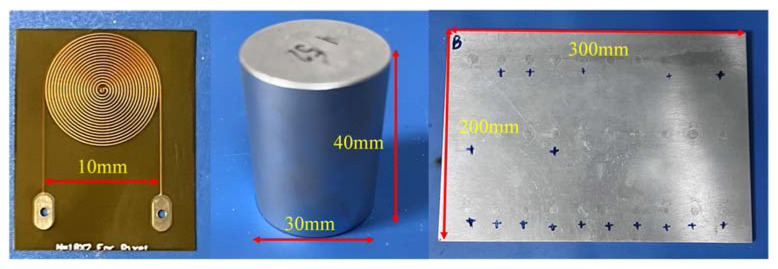
The coil, magnet, and CHRJ sample.

**Figure 9 materials-16-03998-f009:**
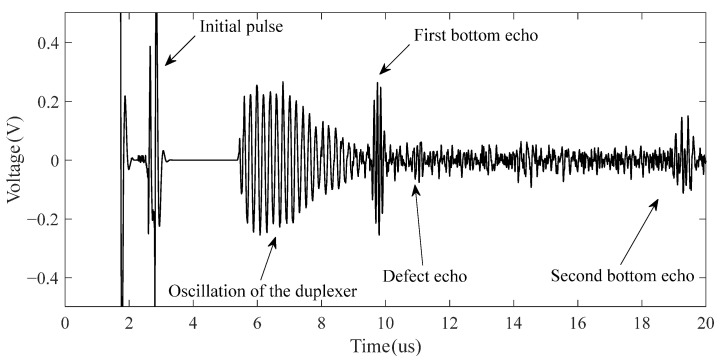
Signal after pre-processing by high-pass filter.

**Figure 10 materials-16-03998-f010:**
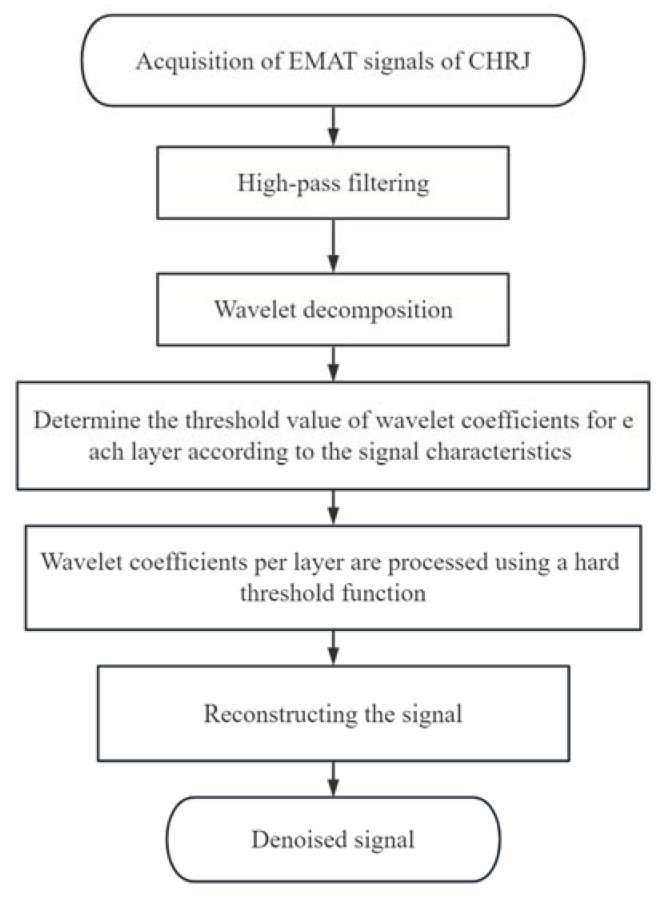
Denoising process of EMAT signals.

**Figure 11 materials-16-03998-f011:**
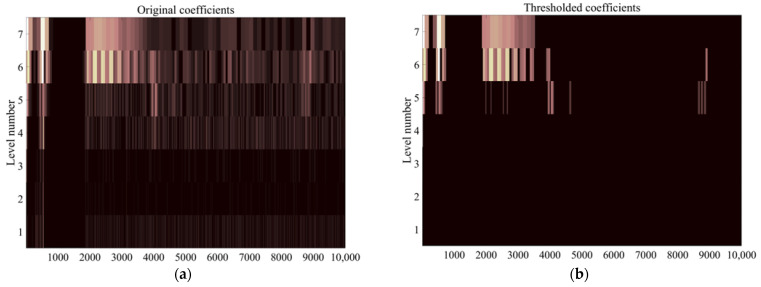
Wavelet coefficients. (**a**) Wavelet coefficients of the pre-processed signal; (**b**) wavelet coefficients after threshold denoising.

**Figure 12 materials-16-03998-f012:**
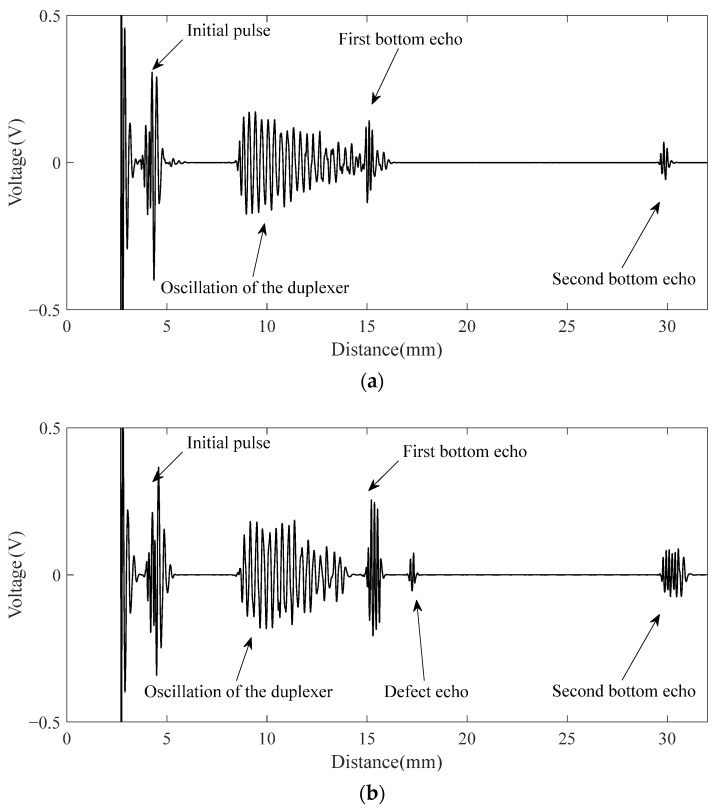
Denoised signals. (**a**) Defect-free CHRJ; (**b**) 0.5-mm-defect CHRJ.

**Figure 13 materials-16-03998-f013:**
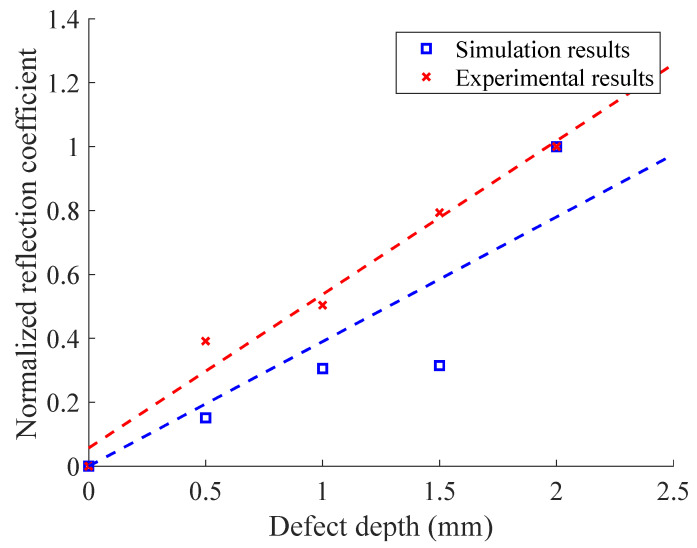
Relationship between simulation and experimental reflection coefficient and defect depth.

**Table 1 materials-16-03998-t001:** Definitions of the variables of Figure 2 and Equation (1).

Name	Meaning
*A_De_*	Amplitude of defect echo
*A_Fbe_*	Amplitude of the first bottom echo
*E*	Total energy of the ultrasonic waves
*r*	Reflection coefficient
*t*	Transmission coefficient
*a*	Energy-loss coefficient

**Table 2 materials-16-03998-t002:** Definitions of the variables in Equation (2).

Name	Meaning
*d_ax_*	Axial resolution
*c*	Wave speed
*n*	Cycles of the ultrasonic wave
*f*	Frequency
*h*	Distance between the defect and the upper surface

**Table 3 materials-16-03998-t003:** Definitions of the variables in Equation (3).

Name	Meaning
*F_lor_*	Lorenz force in the specimen
*J_ind_*	Current density of induced eddy currents in the specimen
*B_static_*	Magnetic flux intensity of static magnetic field
*J_re_*	Current density in the conductor during reception
*η*	Conductivity of the specimen
*v*	Velocity of the particle in the specimen

**Table 4 materials-16-03998-t004:** Experimental setup parameters.

Object	Name	Value
RITEC RPR4000	Frequency	10 MHz
	Cycles	3
	Excitement peak voltage	500 V
	Gain	72.4 dB
	Passband of filter	800 kHz–22 MHz
Oscilloscope	Sampling rate	100 MHz
	Average number of samples	256
	Data-acquisition resolution	10 ns

**Table 5 materials-16-03998-t005:** Parameters of EMAT.

Object	Name	Value
Cylindrical magnet	Height	40 mm
	Radius	15 mm
	Material	N52
	Residual magnetic flux density	1.48 T
	Distance from magnet to coil	0.1 mm
Spiral coil	Spacing	10 mil
	Line width	5 mil
	Number of turns	18
	Radius	5 mm
	Copper thickness	0.5 oz
	Number of layers	2
	Lift-off	0.02 mm

## Data Availability

The data supporting the results reported by the authors can be sent by e-mail.

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
