# Peer review of "Near-Surface-Defect Detection in Countersunk Head Riveted Joints Based on High-Frequency EMAT"

_materials, 2023, doi:10.3390/ma16113998_

Round 1

Reviewer 1 Report

Dear Author(s);

 In this article, the following points need to be solved:

 1.      The numerical data given in the article should be given in tabular form. Not in the text.

2.     At the end of the introduction, the originality of the study should be emphasized in the form of items.

3.     Photograph of the CHRJ should be added to Fig.1.

4.     The values and definitions of the variables in the given equations for this study should be given in tables.

5.     Section 4.1 of the article is very inadequate. For example, the system's electronics, data acquisition program interface are not given. Moreover, information such as SNR of the system, data acquisition resolution is also missing.

6.     Fig.9 and Fig.11 should be explained in detail. How was the naming of the original signals in Fig.9 done? For example, why is the signal change between the 14th and 16th seconds not given a name? In Fig.11, how was the threshold coefficient determined?

7.     Conclusion section should be rewritten. The results should be compared with the literature in the table. It should not be like a summary of study.

Result: Major revision

-

Reviewer 2 Report

This paper studied the detection of near–surface defect in the CHRJ based on high frequency electromagnetic acoustic transducers (EMAT). The working is interesting and well conducted, I just got few comments:

In the Introduction it would be helpful to briefly include some literatures about stress concentration of joints, e.g. 10.1016/j.compositesb.2015.07.018; 10.1016/j.engstruct.2015.05.033

However, none of the above detection methods can effectively detect defects in rivets. How did you get this conclusion? If this is the case, then maybe it is not appropriate to introduce them here. Also, what does ‘effectively detect defects’ mean?

Line 50, EMAT is unclear here, the full name should be given although it has been given in the abstract.

‘et al.’ is missing in almost all of the cited literatures which have more than 2 authors.

The literatures review is sufficient, but all of them are too specific, with loads of detailed research content. This make it not easy to follow and not quite clear what is the research gap, I suggest use your own language to summarise all the literatures and emphasize the research gap a bit more.

Add a scale bar in Fig. 8, or add the main dimensions of the sample.

‘Results and discussion’ is a bit too short, so is ‘Conclusions and Future Works’

Reviewer 3 Report

The article presents a non-contact method of detecting defects in countersunk head rivet joints. While Near-surface defect detection with EMATs is not novel, the use of wavelet post-processing to improve SNR with high frequency EMAT signals is somewhat novel. Additionally, the application of using transverse waves for near-surface defect detection in CHRJ also can be considered somewhat novel. The authors present a good correlation with defect after denoising. 

There are some important questions  that still need to be answered before the work can be considered for publication.

1. The location of the scan relative to the rivet being inspected should be included. Was there movement during the scan?

2. How is the EMAT spaced from the surface under inspection? What material and distance? Also, what is the magnetic field strength of the magnet? If someone wants to replicate this experiment, I don’t think it can be currently done without more details. 

3. Corrosion is common in the defect areas studied in this article, how does this method ensure that the detected defect is not corrosion of the aluminum plates?

4. Are the dimensions used for the simulations the same used in the experiments?

5. Add units to figure 4. Figure 5 arrows could be thicker to see.

6. Figure 7 could be replaced with a schematic of the experimental setup. More details, such as EMAT location and coil lift-off should be included here.

 7. 86-87: near surface CHRJ defects are best detected with high frequency transverse EMAT, why?

8. 87-89: Better explain what the ultrasonic blind zone would make this more coherent.

9. 198: no Results and Discussion section as stated

 10. Fig. 13: Can you force the fit equation to go through 0? a negative reflection coefficient in this context does not mean anything. 

The manuscript can benefit from a close edit for language. 

Reviewer 4 Report

This paper presents a method to detect cracks in rivets. The method uses an EMAT and signal processing to measure the second reflection on the defect.

The method used is clearly explained, however, some information need to be added to the paper before it can be published:

- What is the size of the defect? this is relevant at multiple places but never mentioned, only its depth s mentioned. For example, from line 140 to 147, the authors explained how they choose the frequency to meet the resolution requirement. This is great except that the requirement is never mentioned, so there is no way to say if it is met.

- For the modeling of the defect, how is the defect simulated? is it a gap in the mesh with the same model as the rivet/plate interface? a nonlinear model? There is no detail on this and, again no mention of the defect size.

- For the experiment, how where the defect generated and how were the depths measured and with what accuracy?

- Finally, in the theory section, I am puzzled by figure 2 and the accompanying text. The authors put an index on the transmission and reflection coefficients and add attenuation only on one path. The transmission and reflection coefficients do not depend on how many times the wave passed through the defect, they are constants, unless the defect is changing during that time, which is not the case here. The amplitude varies depending on the distance traveled by the wave due to the attenuation (also constant). The authors seem to account for attenuation only below the rivet and then bundle its effect within variable transmission and reflection coefficients. This does not change the analysis based on the ratio of received amplitudes, but the explanation is at best confusing and at worst represent a lack of understanding of basic wave propagation. This needs to be fixed.

Based on those comments, I cannot recommend publishing this paper in its current state, but it could be reconsidered after these comments are addressed.

Round 2

Reviewer 1 Report

Congratulations to the authors.

Congratulations to the authors.

Author Response

Thank you for your review.

Reviewer 4 Report

My comments have been addressed. The only concern I have before publication is one of the addition at the end of section 4 (lines 299-301) that states "The main reason is that the transverse waves used in this paper are longitudinal waves, which cause more energy to leak into the aluminum plate".

I assume the author means that while the study in ref [25] uses longitudinal waves, the current study uses transversal waves. I am guessing this is an English issue and would suggest a rework of that sentence before publication.

see my comment above: that sentence is non sensical but likely just an English issue.

Author Response

We have revised the expression here, thank you for your review.